# Impact of Different Light Conditions on the Nitrogen, Protein, Colour, Total Phenolic Content and Amino Acid Profiles of Cultured *Palmaria palmata*

**DOI:** 10.3390/foods12213940

**Published:** 2023-10-27

**Authors:** Anthony Temitope Idowu, Miryam Amigo-Benavent, Susan Whelan, Maeve D. Edwards, Richard J. FitzGerald

**Affiliations:** 1Proteins and Peptides Research Group, Department of Biological Sciences, University of Limerick, V94 T9PX Limerick, Ireland; anthony.t.idowu@ul.ie (A.T.I.); miryam.amigobenavent@ul.ie (M.A.-B.); 2BioMaterial Research Cluster, Bernal Institute, University of Limerick, V94 T9PX Limerick, Ireland; 3Health Research Institute, University of Limerick, V94 T9PX Limerick, Ireland; 4Irish Seaweed Consultancy Ltd., H91 TK33 Galway, Ireland; susan.whelan@irishseaweed.com (S.W.); maeve.edwards@irishseaweed.com (M.D.E.)

**Keywords:** amino acid, light regime, nitrogen, *Palmaria palmata*, protein, red seaweed

## Abstract

The impact of different light conditions during culture on the nitrogen, protein, colour, total phenolic content (TPC) and amino acid profile of *Palmaria palmata* biomass was investigated. *P. palmata* was cultured using different light regimes, i.e., white (1 and 2), red, blue and green over 12 days. A significant decrease (*p* < 0.05) in total nitrogen (TN), non-protein nitrogen (NPN) and protein nitrogen (PN) was observed on day 6 while an increase was observed on day 12 in *P. palmata* samples cultured under blue light. The protein content (nitrogen conversion factor of 4.7) of the initial sample on day 0 was 15.0% (*w*/*w*) dw whereas a maximum protein content of 16.7% (*w*/*w*) was obtained during exposure to blue light following 12 days culture, corresponding to an 11.2% increase in protein content. Electrophoretic along with amino acid profile and score analyses showed light-related changes in protein composition. The lighting regime used during culture also influenced the colour parameters (lightness L*, redness a*, yellowness b* and colour difference ΔE) of milled algal biomass along with the TPC. Judicious selection of lighting regime during culture may allow the targeted production of sustainable high-quality proteins from *P. palmata.*

## 1. Introduction

Global consumer protein demand is expected to continue growing due to an increasing human population which is predicted to reach over 9 billion in 2050 [1,2]. In 2019, the global protein ingredient market was valued at 38.02 billion USD with a proposed total growth of 9.1% between the period of 2020 to 2027 [2]. In addition, it has been projected that the global protein demand would likely increase to 943.5 million metric tons by 2054, indicating a need for the development of more sustainable protein ingredients, including plant-based proteins for human consumption [3]. Macroalgae, currently consumed mostly by Asian populations, has been identified as a nutrient-dense food having a high protein content making it a potentially good source of alternative protein [4]. Different macroalgal species contain protein levels that are equivalent to those found in common protein-rich foods such as cereals, soybeans, fish and eggs [5]. The protein quality and digestibility of macroalgal proteins have been reported to be on par with traditional protein sources [6,7]. Hence, macroalgal protein may find applications in the development of protein-rich food ingredients.

*Palmaria palmata*, also known as dulse, is a red macroalga that is widely distributed along the North Atlantic coastlines within a latitude of 40–80° N [8]. It has an established economic value as it has been consumed as a food for several centuries [9]. Furthermore, *Palmaria palmata* is one of the few macroalgae species approved by the European Union’s novel food catalogue [10]. It is usually hand-picked from the seashore, alternatively, it can be cultured in tanks or on longlines when required for larger-scale applications [11,12,13]. The harvest from both wild and cultured *P. palmata* in, e.g., Ireland is currently estimated to be approximately 100 tonnes per annum [11,14,15].

The literature in relation to *P. palmata* cultivation is generally focused on biomass yield improvement by enriching the nutrient composition of the seawater, controlling the temperature, changing the irradiance or intensity of light, and/or the salinity of water when grown in aquaculture systems [16]. The irradiance or intensity of light along with the spectral composition and duration of light exposure is critical for seaweed growth because these parameters are known to impact the rate of photosynthesis, the structural distribution and the cellular composition (e.g., protein and lipid contents) of seaweed [17,18,19,20,21]. Light irradiance may therefore impact metabolic regulation, protein expression, reproduction and ultimately macroalgal growth [22,23]

In general, lighting such as that obtained from light-emitting diodes (LED), fluorescent, incandescent and sunlight sources can be used for the culture of seaweed [24,25]. Choi et al. [26] stated that LEDs could be preferred to fluorescent and incandescent lamps or bulbs for seaweed biomass culture because of their low power consumption, durability and high efficiency of electricity conversion. In addition, LEDs are solid-state semiconductors, which produce light with a narrow emission spectrum (20–30 nm at half peak height). When current is applied to LEDs photons are produced. These photons possess different energies and wavelengths. Therefore, distinct parts within the photosynthetically active region (PAR) can be assessed individually, thereby precisely resolving an organism’s specific photosynthetic response. The main absorption peaks in the PAR depend on the carotenoid and the individual chlorophyll composition of the macroalgae [27]. Furthermore, the drive for sustainability and a reduction in energy consumption among many European countries makes the use of LEDs preferable in terms of savings in energy and associated costs for seaweed producers [28]. This makes LEDs ideal light sources for supporting the growth of plants and other photosynthetic organisms (such as macroalgae) when cultured in a controlled environment [18].

In general, studies on increasing the protein content of *P. palmata* using different light regimes are limited. Parjikolaei et al. [29] studied the effect of light colour (red, white and blue) at the same intensity (55 µmol photons s^−1^ m^−2^) and nitrogen nutrient source (30 and 440 µM nitrate) made up of nitrate, phosphate, trace metal and vitamin stock solutions on biomass yield and pigment content (β-carotene) in *P. palmata* cultured over 4 weeks. The growth rate increased significantly in *P. palmata* cultured with white light (90%), followed by red light (45%) and blue light (40%) when a nutrient concentration of 440 µM was maintained, respectively. When a nitrate concentration of 30 µM was maintained, the biomass yield of cultured *P. palmata* increased in white light (40%), followed by red light (30%) and blue light (25%), respectively. Overall, the highest pigment composition (β-carotene) was observed when *P. palmata* was cultured under white light with a nitrate concentration of 440 µM. The study indicated that the culture conditions impacted the growth rate, the biomass yield and the pigment content. However, the nitrogen, protein, total phenolic content and amino acid profile of the biomass obtained from the different light treatments were not studied. The use of different lighting treatments during the culture of biomass from other red macroalgal species has been reported. For instance, Barufi et al. [30] investigated the effects of different spectral qualities on the reproduction, growth and pigment contents (including the protein phycoerythrin) of *Gracilaria birdiae*. Tetraspores were cultured using a 40W incandescent Osram light bulb with different light colours: white, green, red and blue over a period of 9 weeks by maintaining a photoperiod (lightness: darkness, L:D) of 14:10 in a biochemical oxygen demand culture chamber. An irradiance of 350 µmol photons m^−2^ s^−1^ was maintained throughout. The growth rates were higher when *Gracilaria birdiae* tetraspores was cultured under white (95%.day^−1^) and green (90%.day^−1^) light at a photoperiod of 14:10. The highest phycoerythrin concentration (~60%) was obtained when *G. birdiae* tetraspores were cultured under blue light. While exposure to white light resulted in a higher growth rate than green light the phycoerythrin content was maximally increased during culture in blue light. Wu [31] reported the effect of different light sources on the growth and photopigment composition of *Pyropia haitanensis*. Fresh thalli of *Pyropia haitanensis* were cultured and exposed to light from different LED sources: red (640–680 nm), blue (420–460 nm), green (490–530 nm) and CK™ fluorescent tube light while maintaining a photoperiod (L:D) of 12:12 over a period of 7 days. Irradiation was maintained at 100 µmol photons m^−2^ s^−1^ for all treatments. The study showed that biomass growth rate increases in *Pyropia haitanensis* when exposed to fluorescent (85%), blue (75%), green (70%) and red (50%) light sources. The phycoerythrin content increased in the following order, i.e., in green (85%), blue (75%), fluorescent (55%) and red (45%) light. Overall, exposure to fluorescent light led to greater increases in the growth rate of *Pyropia haitanensis* while exposure to green light was most favourable for the accumulation of phycoerythrin.

The above studies indicate the possibility of increasing growth rate, biomass and pigment composition on exposure of macroalgae to different light spectra and/or qualities. However, knowledge gaps exist that specifically compare the impact of light from different LED sources on protein content and quality during, e.g., the culture of *P. palmata* as a function of different growth periods. Furthermore, to our knowledge there appears to be no study to date that investigates the impact of light from different LED sources on the nitrogen, protein, colour, total phenolic content (TPC) and amino acid profile of cultured *P. palmata*. Therefore, the objective of this study was to examine the impact of different LED light colours (white, red, blue and green) on chemical composition during the culture of *P. palmata* under controlled laboratory-scale cultivation conditions. Therefore, the impact of different lighting treatments during the culture of *P. palmata* over 12 days on its nitrogen, protein, colour, TPC and amino acid profiles was investigated herein.

## 2. Materials and Methods

### 2.1. Materials

Trichloroacetic acid (TCA), Kjeldahl tablets, 1 mM ethylenediaminetetraacetic acid (EDTA), phenol solution (equilibrated with 10 mM Tris HCl, pH 8.0), 1,4-dithiothreitol (DTT), iodoacetamide, Folin-Ciocalteau reagent, sodium carbonate, gallic acid and wide range molecular weight markers (6.5 to 200 kDa) were supplied by Sigma (Dublin, Ireland). Hydrochloric acid (HCl) and sodium hydroxide (NaOH) were procured from VWR (Dublin, Ireland). Ethanol was supplied by ThermoFisher Scientific (Dublin, Ireland). Protein Loading Buffer Blue (2×) and Tris-glycine sodium dodecyl sulphate polyacrylamide gel electrophoresis (SDS-PAGE) buffer (10×) were obtained from National Diagnostics (Atlanta, GA, USA). Mini-protean TGX stain-free 8–16% gradient gels were obtained from Bio-Rad (Hercules, CA, USA). Bicinchoninic acid (BCA) protein assay kits were supplied by ThermoFisher Scientific (Dublin, Ireland). All other reagents were of analytical grade.

### 2.2. Vegetative Culture of P. palmata with Different LED Light Sources

*P. palmata* biomass was widely harvested from Moyrus, Co. Galway, Ireland (53.3663, −9.8930). Prior to the culture experiments, this biomass was acclimated under laboratory conditions for 3 weeks. Young, clean apical tissue was excised from the *P. palmata* fronds after cleaning by rinsing with cartridge- and UV-filtered seawater (filtered to 1µm) and further acclimated in a 20 L tank with UV-filtered seawater for another 7 days. Acclimation parameters were a 12:12 Light:Dark (L:D) photoperiod at 60 μmol m^2^ s^−1^ provided by two white LED light sources (Spectron, Light Science, 20 W 1.2 GB cool white, Amsterdam, The Netherlands). The seawater was held at 11 ± 1 °C and constantly aerated by agitation to allow the tips to circulate freely. Nutrients were not added to the seawater during the acclimation period. Following acclimation, the trials were carried out in 5 L glass bottles (*n* = 3). Culture conditions were as follows: using a *P. palmata* tank stocking density of 4 g L^−1^ and a 12:12 L:D photoperiod at 100 μmol m^−2^ s^−1^ PPFD (Photosynthetic Photon Flux Density, 400–780 nm) measured using a LI-COR spectrometer (model LI-180, LI-COR Biosciences, Germany). LED luminaires (Philips GreenPower LED Production Module 3.0 Dynamic model, supplied from Signify, Gulliford, UK) coupled with a Philips Growise Control System (GWCS) (Signify, UK) provided sources of red, blue, green and white (white 1) light. The GCWS software (442294406021 E) was used to programme a “light recipe” labelled as “white 1”, blending white and coloured LEDs that approximated the spectral composition and intensity of a second LED source (Spectron, Light Science, 20 W 1.2 GB cool white, Amsterdam, The Netherlands) which was labelled as “white 2” light. Measurements of the spectral compositions of all LED light sources were analysed with a handheld spectrometer (model LI-180, LI-COR Biosciences, Walsmuhler, Germany). Agitation with aeration was maintained to ensure biomass circulated freely in the tank while the seawater temperature was maintained at 11 ± 1 °C. Nutrients in the form of F/2 medium (Cell Hi F2P, Varicon Aqua Solutions, Hallow, UK) at 0.1 g L^−1^ were incorporated into the seawater on day 0 and subsequently replenished after the seawater change which was conductedn on day 6. All trials were conducted for a total of 12 days. The position of the glass bottles in the light-controlled unit was changed every 3 days to ensure homogenisation of the lighting conditions. Blotted fresh-weight biomass was weighed every 3 days and excess biomass was removed on day 6 to ensure that the optimal stocking density (4 g L^−1^) was maintained in the tanks during the experimental period. Biomass samples were rinsed in filtered seawater prior to N fraction determination to remove potential interference from residual nutrient medium. Thereafter, samples were freeze-dried (Labconco FreeZone12, MI, USA), milled to an average size of <3 mm (Krups F203 Grinder, MI, USA) and then labelled after the different light exposures, i.e., white 1, red, blue, green and white 2, respectively. Dry biomass was stored at room temperature in air-tight containers for subsequent nitrogen, colour, TPC and AA profile analyses.

### 2.3. Determination of Total Nitrogen (TN), Protein Nitrogen (PN) and Non-Protein Nitrogen (NPN)

The TN content of the *P. palmata* samples obtained following different colour light exposures was analysed using the macro-Kjeldahl protocol that was described by Connolly et al. [32]. The PN and NPN content of the macroalgal samples cultured under different light treatments and obtained on days 6 and 12 were analysed using the protocol described by Stack et al. [33]. The results obtained were expressed in terms of percentage (g nitrogen/100 g dw biomass). All samples were analysed in triplicate.

### 2.4. Colour Measurement

Colour parameters (L*, a*, b*) for milled *P. palmata* samples were analysed using a Konica Minolta CR-400 Chroma Meter (Minolta Camera Co., Osaka, Japan) as described by Cermeño et al. [34]. The colour difference between the initial samples and those cultured under different light was analysed as previously described by Cermeno et al. [35] and recently modified by Idowu et al. [7].

### 2.5. Total Phenolic Content (TPC)

Phenolic compounds were extracted using the protocol described by Amigo-Benavent et al. [36] with minor modifications. Solvents containing 50% (*v*/*v*) methanol and 70% (*v*/*v*) acetone were used to extract liquid phenolic compounds from the cultured *P. palmata* samples. The liquid extract was used for the determination of the TPC content using the Folin-Ciocalteau (F-C) assay [37,38].

TPC analysis of *P. palmata* samples was carried out following the protocol of Connolly et al. [31]. The absorbance was read at 765 nm (Shimadzu UV-1800, Kyoto, Japan). A standard calibration curve using gallic acid was prepared in the same way as the samples at a range of concentrations from 5–80 µg/mL. The TPC values were reported as mg gallic acid equivalents (GAE)/g dw of the sample. This experiment was conducted and analysed in triplicate.

### 2.6. Protein Extraction and Quantification for Gel Electrophoretic Analysis

Protein was extracted from the *P. palmata* samples following the protocol outlined by Wang et al. [39]. The protein content of the extracted sample was quantified using the BCA assay as previously described by Idowu et al. [7].

### 2.7. Sodium Dodecyl Sulfate Polyacrylamide Gel Electrophoresis (SDS-PAGE)

SDS-PAGE was performed on the *P. palmata* protein extracts obtained from the biomass collected after 6 and 12 days of incubation under different light treatments following the procedure described by Idowu et al. [7].

Relative quantification of the protein bands in the stain-free SDS-PAGE gels of the *P. palmata* protein extracts after 6 and 12 days of culture was carried out by densitometry using a Uvi-Tec imaging system and UViBand Essential v6 Software (Uvi-TEC). The relative protein content in individual protein bands was expressed as the percentage volume of each band in relation to the total volume of the initial sample on day 0.
(1)$$Protein\;band\;volume\;(\%)=\frac{Volume\;of\;individual\;protein\;band}{Total\;volume\;of\;bands\;for\;the\;intial\;sample\;}$$

### 2.8. Amino Acid (AA) Profile

AA analysis was performed on the *P. palmata* samples obtained following different light exposure treatments (the day 6 sample cultured under blue light could not be analysed due to limited sample availability). Analysis was carried out using the external accredited service of the CIB Margarita Salas (CSIC, Madrid, Spain) following the procedure outlined by Friedman [40]. The results were expressed as g of AA/100 g dw.

The amino acid score (AAS) of the *P. palmata* samples obtained from the different lighting treatments was calculated as mg of AA per g of protein divided by the mg of the same amino acid per g of the reference protein for the essential amino acids [41,42]. Protein content in *P. palmata* was obtained using the amount of protein nitrogen (*w*/*w*) in the test samples multiplied by a nitrogen conversion factor of 4.7 following Bjarnadóttir et al. [43]. The reference protein was based on the AASs provided by the Food and Agriculture Organization (FAO) where the reference protein requirements are indicated for two population cohorts, i.e., (1) children (from 6 to 36 months old) and (2) older children (older than 36 months), adolescents and adults [41].

### 2.9. Statistical Analysis

Data were statistically analysed by one-way analysis of variance (ANOVA) with Tukey post hoc comparison to test at a significance level of *p* = 0.05, where applicable on SPSS (Version 26, IBM Inc., Chicago, IL, USA). Correlation, if any, between TN, PN and NPN with ΔE as well as TPC was performed using Pearson’s correlation test on SPSS.

## 3. Results

### 3.1. Nitrogen Profiles of the P. palmata Samples Cultured under Different Light Treatments

The TN is the sum of the PN and NPN and can be obtained by direct nitrogen quantification of algal samples using the Kjeldahl method [44]. The PN reflects the true nitrogenous constituents that are proteins, they are obtained as a pellet following algal sample precipitation with trichloroacetic acid (TCA) at a concentration of 12% (*w*/*v*). The supernatant obtained following the TCA precipitation is the NPN. These nitrogenous constituents are then quantified using the Kjeldahl method [33,45]. The effect of LED colour lighting during culture on TN content was assessed for *P.palmata* samples, which were subsequently grown for 6 and 12 days under similar laboratory growth conditions except using different light regimes. The TN content ranged between 3.47 ± 0.03 and 4.73 ± 0.05% for the biomass obtained when grown under different light treatments while the starting sample (day 0) contained a TN content of 4.56 ± 0.04% (Table 1). Statistically significant differences (*p* < 0.05) were observed in the TN content as a function of the different light regimes and growth period. The highest TN content was observed in *P. palmata* cultured under blue light and was obtained on day 12 (4.73 ± 0.05%) while the lowest TN content was observed in *P. palmata* cultured under red light on day 6 (3.47 ± 0.03%). There was an overall significant decrease in TN for all light trial samples obtained on day 6 in comparison to the initial sample. i.e., initial sample (4.56 ± 0.04%), white I (3.75 ± 0.05%), red (3.47 ± 0.0 3%), blue (3.71 ± 0.04%), green (3.83 ± 0.05%) and white 2 (3.49 ± 0.05%) (*p* < 0.05). Thereafter, the TN content significantly increased after 12 days of culture under blue light (4.73 ± 0.05%) in comparison with the initial sample (4.56 ± 0.04%). Culture under the other light treatments did not increase the TN content in comparison with the initial sample (Table 1). All samples displayed increases in their TN content after 12 days of culture in comparison with the same sample grown for six days.

The PN values of the *P. palmata* cultured with different light treatments decreased significantly (*p* < 0.05) in comparison to the initial sample on day 6, i.e., initial sample (3.20 ± 0.03%), white 1 (2.60 ± 0.04%), red (2.36 ± 0.06%), blue (2.48 ± 0.00%), green (2.72 ± 0.07%) and white 2 (2.42 ± 0.06%). Similar PN values were obtained for white 1 (3.06 ± 0.06%) and green (3.15 ± 0.05%) light-cultured samples on day 12 (*p* > 0.05) in comparison with the initial sample. The highest PN was obtained in cultured *P. palmata* exposed to blue light (3.56 ± 0.06%) obtained on day 12.

The NPN values of all *P. palmata* samples cultured under different colour lights decreased significantly herein in comparison to the initial sample on day 0 (1.21 ± 0.07%). Similar NPN values were obtained for the cultured samples on day 6 (NPN values ranged between 0.93 ± 0.01 to 1.02 ± 0.06%) and on day 12 (NPN values ranged between 0.95 ± 0.07 to 1.01 ± 0.01%).

### 3.2. Colour Parameters of Cultured P. palmata

The effect of light spectra on *P. palmata* during culture was assessed in terms of the colour of the biomass produced. Figure 1A–C shows the colour parameter (L*, a* and b*) values for the milled cultured biomass. It was observed that the L* values decreased (indicating darker colour) in samples exposed to different light treatments and obtained on day 6 (ranging from 27.90 ± 1.21 to 31.19 ± 2.00) and on day 12 (ranging from 27.91 ± 0.56–29.43 ± 1.02) in comparison with the initial sample obtained on day 0 (34.28 ± 0.89) which exhibited the highest L* value. An increase in positive a* value (indicating more redness) was observed for samples obtained on day 6 (7.49 ± 0.56–8.07 ± 0.31) and on day 12 (7.23 ± 0.27–8.04 ± 0.27) in comparison to the initial sample on day 0 (6.83 ± 0.33). The highest a* value was observed in the red light cultured sample (8.94 ± 0.92) obtained on day 6. An increase in b* values was observed in the sample, i.e., white 1 (6.55 ± 0.35), red (8.33 ± 1.36), green (7.46 ± 0.51) and white 2 (6.87 ± 0.48) following six days culture. Similar results were also obtained for white 1 (6.09 ± 0.32), red (7.99 ± 0.11), green (6.71 ± 0.35) and white 2 (6.99 ± 0.12) light-exposed samples obtained on day 12. Furthermore, similar values were observed for the initial sample obtained on day 0 (5.56 ± 0.24) and the sample exposed to blue light for both day 6 (5.57 ± 0.58) and day 12 (5.37 ± 0.08). The highest b* value (indicating more yellowness) was observed in the red light cultured sample (8.33 ± 1.36) obtained on day 6. The colour difference (ΔE) values of samples following exposure to different light treatments were also calculated (Figure 1D). The ΔE increased from 3.16 ± 1.18 for the blue light cultured sample obtained on day 6 to 6.92 ± 0.63 for the red light cultured sample obtained on day 12. Similar ΔE values were observed between white 1 (5.94 ± 0.47), red (6.43 ± 1.30) and white 2 (6.63 ± 0.40) samples obtained on day 6 with white (6.17 ± 0.17), red (6.92 ± 0.63) light treatment samples obtained on day 12. Furthermore, a similar ΔE value was observed between green (5.00 ± 0.32) and white 2 (5.38 ± 1.49) light-cultured samples both obtained on day 12. The largest change in colour of 6.92 ± 0.63 was observed in the red light cultured sample obtained on day 12.

### 3.3. TPC of P. palmata Samples Cultured with Different Light Regimes

The TPC of the biomass obtained from *P. palmata* that was cultured under different light spectra was quantified using the Folin Ciocalteau assay. The TPC ranged between 4.41 ± 0.33 mg GAE/g dw for *P. palmata* cultured under green light and obtained on day 6 to 11.21 ± 0.07 mg GAE/g dw for the sample cultured with red light obtained on day 6 (*p* < 0.05) (Figure 2). The initial sample had a TPC of 7.32 ± 0.11 mg GAE/g dw. Overall, different TPCs were observed for the samples cultured under the same light regime as a function of culture duration, e.g., *P. palmata* samples cultured under white 1 showed a statistically significant increase in TPC on day 12 (7.92 ± 0.04 mg GAE/g dw) in comparison to day 6 (6.01 ± 0.23 mg GAE/g dw). The blue light culture sample obtained on day 12 had a TPC of 5.91 ± 0.06 mg GAE/g dw. The samples cultured under blue light obtained on day 6 were not analysed due to sample availability limitations. There was no significant different between the TPC of green light cultured samples obtained on day 6 versus white 2 culture obtained on day 12, i.e., 4.41 ± 0.03 vs. 4.83 ± 0.15 (*p* > 0.05). The white 1 light cultured sample obtained on day 6, green light cultured sample obtained on day 12, blue light cultured sample obtained on day 12 and white 2 light cultured sample obtained on day 6 showed no significant differences, with similar TPC’s of 6.01 ± 0.23, 6.27 ± 0.18, 5.91 ± 0.06 and 6.14 ± 0.41 mg GAE/g dw, respectively (*p* > 0.05).

### 3.4. Correlation between TN, PN, NPN and ΔE

Pearson’s correlation test was used to assess the relationship, if any, between the TN of *P. palmata* samples cultured with different light sources and the ΔE (Appendix A). The analysis indicated a negative correlation (r = −0.230, *p* = 0.522) which was not significant. This test was also used to assess the relationship, if any, between PN and the ΔE (r = −0.125, *p* = 0.731) as shown in Appendix A.

Pearson’s correlation test was also used to assess the relationship between NPN and ΔE (r = −0.202, *p* = 0.576). The Pearson correlation analysis between PN and NPN was also assessed. A positive correlation existed between PN and NPN (r = 0.195, *p* = 0.589). Furthermore, Pearson’s correlation test showed a negative correlation between TN and TPC (r = −0.211, *p* = 0.558), PN and TPC (r = −0.246, *p* = 0.494) and a non-statistically positive correlation between NPN and TPC (r = 0.209, *p* = 0.563) in Appendix A.

### 3.5. SDS-PAGE Analysis

The protein profiles of cultured *P. palmata* obtained using SDS-PAGE are shown in Figure 3.

Visible and clear bands (labelled 1–11) with molecular masses corresponding to 87, 71, 58, 53, 41, 34, 29, 23, 20, 18 and 15 kDa were observed. The intensity of band 11 (15 kDa) was more visible specifically in lanes a, b, and c which correspond to the initial sample (day 0), white 1 (day 6) and red (day 6) samples in comparison to other samples obtained on day 6 and day 12.

The relative quantitative distribution of the protein bands of *P. palmata* cultured using different light systems and obtained after 6 and 12 days of growth was analysed using densitometry (Figure 4A,B). The result obtained indicated differences in protein expression as a function of the different light spectra used during culture and the duration of growth. The % relative band volume for bands 3, 9, 10 and 11 was generally high for both the initial sample and for samples cultured under different light regimes which corresponded to a high intensity in the overall protein profile (Figure 4A,B).

### 3.6. AA Profiles

The AA composition *P. palmata* biomass obtained following culture under different light regimes is summarised in Table 2. The essential AA (EAA) content ranged between 8.26 ± 0.10 (g/100 g dry weight) for samples cultured in white light 2 and grown for 12 days to 12.71 ± 3.88 (g/100 g dry weight) for the initial sample. However, there was no difference between the EAA of the initial sample and the sample cultured under blue light which was 12.51 ± 0.58 (g/100 g dry weight) and the sample cultured under green light which was 12.06 ± 0.81 (g/100 g dry weight) after 12 days growth on a dry weight basis (*p* > 0.05). In addition, no statistically significant difference existed between the EAA of samples cultured under different treatments obtained on day 6, i.e., white 1 (10.11 ± 2.66, g/100 g dry weight), red (8.92 ± 1.30, g/100 g dry weight), green (10.33 ± 0.77, g/100 g dry weight), white 2 (9.23 ± 1.33, g/100 g dry weight) (*p* > 0.05). Meanwhile, significant differences occurred in the EAA of samples cultured under different light systems obtained on day 12., i.e., white 1 (11.11 ± 0.10, g/100 g dry weight), red (9.29 ± 0.64, g/100 g dry weight), blue (12.51 ± 0.58, g/100 g dry weight), green (12.06 ± 0.81, g/100 g dry weight) and white 2 (8.26 ± 0.10, g/100 g dry weight) (*p* < 0.05).

The non-essential AA (NEAA) content ranged between 13.53 ± 1.63 (g/100 g dw) for the sample cultured under green light and grown for 6 days to 16.97 ± 1.19 (g/100 g dw) for the sample cultured under white 1 light and grown for 12 days. The study herein showed significant (*p* < 0.05) increases in the NEAA content of *P. palmata* cultured with different light regimes specifically in white 1 (16.97 ± 1.19, g/100 g dw), blue (15.47 ± 2.04, g/100 g dw), white 2 (15.34 ± 0.85, g/100 g dw) and green (15.30 ± 1.37, g/100 g dw) light after 12 days when compared to the initial sample (14.94 ± 3.97, g/100 g dw) (Table 2). The branched-chain amino acid (BCAA) content herein ranged between 2.75 ± 0.34 g/100 g dw for the biomass cultured under white 2 (day 6) to 5.10 ± 0.13 g/100 g dw for the biomass cultured under blue light (day 12) sample. There were increases in BCAA content in *P. palmata* cultured under blue light (5.10 ± 0.13, g/100 g dw) and green (5.01 ± 0.34, g/100 g dw) light obtained on day 12 in comparison with the initial sample (4.79 ± 1.56, g/100 g dw). The BCAA of cultured *P. palmata* under different light regimes obtained at day 6 decreases significantly in comparison with the initial sample, i.e., white 1 (4.01 ± 1.06, g/100 g dw), red (3.39 ± 0.52, g/100 g dw), green (3.11 ± 0.46, g/100 g dw) and white 2 (2.75 ± 0.34, g/100 g dw) light in comparison to the initial sample (4.79 ± 1.56, g/100 g dw) (*p* > 0.05). The overall TAA ranged between 22.66 ± 3.16 g/100 g dw for the biomass cultured under red light obtained on day 6 to 27.98 ± 2.61 g/100 g dw for the biomass cultured under blue light obtained on day 12.

The amino acid score (AAS), an indication of protein quality, of the *P. palmata* biomass obtained following culture under different light regimes is shown in Figure 5. The calculated amino acid scores ranged from 1.34 to 14.49 and 1.43 to 17.00 per residue when referenced against the recommended amino acid requirement for the two selected population cohorts, i.e., (i) children from 6 months to 3 years and (ii) in the older children (older than 3 years old), adolescents and adult population cohorts, respectively [41].

## 4. Discussion

### 4.1. Nitrogen Profiles of the P. palmata Samples Cultured under Different Light Treatments

Nitrogen is a pivotal element that is necessary to support the growth of seaweed [46] and plant metabolism [47]. In this study, a decrease in TN content was observed on day 6 under all light regimes which may be due to the time required for photo-acclimatization of the *P. palmata* biomass to accommodate the different light spectra used. Photo-acclimatization can trigger physiological changes in the phycobilisome (light-harvesting antennae) due to the disruption in environmental conditions [48]. Lehmuskero et al. [49] reported the possibility for algae to adjust to variations in lighting conditions leading to alteration in metabolic, physiological and transcriptional activities, which further manifests as a change in the photosynthetic capacity of the cell. However, upon acclimatization, increases in the TN could be observed on day 12. Light is vital for chlorophyll synthesis and macroalgal pigment synthesis is regulated by different photoreceptors that absorb light at different wavelengths [50]. Elsewhere, Figueroa et al. [51] also stated that light plays a vital role in seaweed morphology. According to Kehoe [48], photosynthetic organisms such as algae use their phycobilisome to increase photon capture on exposure to light and provide the energy needed to drive photosynthesis. The result obtained herein indicated the possibility of increasing TN on exposure specifically to blue > green > white 1 > red > white 2 light on day 12. Leukart and Lüning [52] stated that the ability of blue and green light to be advantageous for red macroalgae biomass development was due to the phylogenetic adaptation of the macroalgal accessory pigments to light in these wavelengths. Blue LED light increased the carotenoid (zeaxanthin) content and white light induced phycobiliprotein production in the red microalgae *Rhodella* sp. [53]. Furthermore, changes in the day/night cycle may lead to phototropic organism and algal assimilation and dissimilation of carbohydrates with enzymes such as glycoside hydrolases involved in carbohydrate metabolism [54]. However, these enzymes can be regulated by blue light and are most likely involved in the accumulation of nutrients to support algae growth. Wu [31] also reported that the rate and efficiency of photochemical electron transport (light absorption) in the thallus of the macroalga on exposure to different light spectra may impact nutrient accumulation in seaweed. This may have contributed to the differences in TN observed in cultured *P. palmata* with different light regimes. López–Figueroa and Niell [55] also reported on the ability of blue light to induce chlorophyll synthesis in the red macroalgae, *Corallina elongata*. In another study, fresh thalli of *Porphyra umbilicalis* were cultured for four weeks at a 12:12 photoperiod (L:D) and under a light intensity of 50 µmol^−2^ s^−I^ in either blue (440–480 nm) or red (630–670 nm) light emanating from two General Electric 20 W fluorescent lamps [51]. The TN composition of thalli grown under blue and red light was 5.00 and 2.65% (dw), respectively. This was attributed to a better acclimatization of the *P. umbilicalis* thalli under blue light which stimulated the accumulation of the photosynthetic pigments. Tsekos et al. [56] investigated the ultrastructure of the vegetative gametophyte of *Porphyra leucosticta* grown under red, blue and green light. The thalli of *P. leucosticta* were cultured for four weeks at an L:D of 12:12 and at a light intensity of 50 μmol m^−2^ s^−1^ under either blue, red or green light provided from fluorescent lamps. Well-formed phycobillisomes were observed on electron microscopic analysis of the sample grown under blue light with fewer phycobillisomes being observed when grown under green light while none were observed during cultured under red light. In addition, the density of the phycobilisomes on thylakoid membranes was higher for blue (800 µm^−2^), followed by red (250 µm^−2^) and green (180 µm^−2^) light grown samples. Therefore, it appears that the type of light spectra used contributed to the differences observed in the evolution of the structure and size of the phycobillisome. The evolution of the phycobillisome during the acclimatization processes is critical since it enables algae and other photosynthetic organisms to respond to stimuli by modifying their phenotype due to changes in irradiance or the spectral distribution of light in a given environment. These modifications are manifested in the form of an increase or a decrease in the number of phycobilisomes per cell, the size, shape, and the chromophore and/or the protein composition of the phycobilisome [57,58]. These modifications could account for the increase in TN observed in *P. palmata* samples herein following 12 days of culture under blue light (Table 1). Overall, the variation in TN observed in the *P. palmata* samples on exposure to different light spectra may be connected with the chromatic acclimation of the *P. palmata* samples and the corresponding spectrum distribution of light in the water bodies during culture. Macroalgae can achieve acclimation by adjusting their photosynthetic organelles via changes in the reaction center ratio, i.e., the membrane-bound pigment–protein complexes located within the photosystem which are important in the photochemical conversion of light into chemical energy, or by changes in the relative content of light protective pigments (Marquardt et al. [59]). These responses in turn may contribute to pigment accumulation in macroalgae. This appears to be the first study reporting on the changes in TN content of *P. palmata* cultured using different colour lights at laboratory-scale as a function of growth days, i.e., 6 and 12 days.

TN is sometimes mistakenly used when calculating the protein content in seaweeds without accounting for the NPN constituents therein which is associated with free amino acids, urea and other forms of nitrogen. Therefore, the PN and the NPN were determined herein in the *P. palmata* samples grown under different lighting conditions. As expected, the PN content was higher than the NPN in all analysed samples (Table 1) constituting between 70.24 to 77.90% of the TN. On converting the PN to protein using a nitrogen-to-protein factor of 4.7 [43], the protein content of the initial sample on day 0 was 15.04% dw whereas the maximum protein content was obtained when exposed to blue light during culture for 12 days was 16.73% dw, which corresponded to an 11.25% increase in protein content. Phycobiliproteins are protein pigments that are found in red seaweed and can represent up to 50% of the total protein content of the seaweed [33]. In this case, the exposure of the cultured *P. palmata* to blue light may have enhanced the accumulation of the phycobiliproteins which resulted in an increase in PN. Wiencke and Bischof [60] and Jerlov [61] reported that different light qualities occur within the water body by virtue of the absorption and scattering of light especially with increasing water depth, agitation and/or turbidity. Furthermore, it was stated that the blue-green light wavelengths penetrate deepest into the water body as the shorter and longer wavelengths are more absorbed by the water molecules or scattered by particles. Hence, the photosynthetic pigments of the algae adapt to these different wavebands which impacts on their photosynthetic efficiencies and activities. A previous study by Idowu et al. [7] indicated the possibility of increasing the protein content of *P. palmata* up to ~50% by culturing with a nutrient containing culture medium (F/2) alone or in combination with urea (0.05 or 0.10 g/L) for 18 days. Therefore, different strategies can be followed in order to increase the protein content in *P. palmata*.

The study herein demonstrated that different light spectra may influence the accumulation of urea, amino acids and ammonia compounds which constitute the NPN fraction. The NPN content constituted 27.4% of TN for the initial sample, and between 25.5 to 29.8% of the TN for the samples cultured for six days and between 22.1 to 25.5% of TN for the samples cultured for 12 days (Table 1). No previous information appears to exist in the literature in relation to the impact of different light exposures on the NPN content during the culture of *P. palmata*.

### 4.2. Colour Parameters of Cultured P. palmata

As already outlined, changes in the phycobiliproteins and beta-carotene content in red algae [62] are most likely responsible for the observed change in colour observed herein. Phycobiliproteins are classified into phycoerythrins and phycocyanins which display red and blue colours, respectively [63]. Chlorophyll is a key component of photosynthesis and is involved in light harvesting which is related to the growth and yield of photoautotrophs [46]. Furthermore, chlorophyll content can fluctuate when algae are exposed to different light spectra [31]. Furthermore, culturing red macroalgae under different light spectra can contribute to the accumulation of pigmented phycobiliproteins [64]. This was evident herein, i.e., when *P. palmata* was exposed to different light regimes during culture which resulted in a decrease in L*, an increase in positive a* and an increase in b* (except in blue light treatment obtained at both day 6 and 12). These findings are associated with an increase in the synthesis of phycobiliproteins which in turn leads to an accumulation of nitrogen in *P. palmata* biomass specifically following 12 days culture.

The findings of the study herein differ from those observed by Vasconcelos et al. [65] who reported on the colour parameters of *P. palmata* fronds grown in tanks at 3 °C with nutrient medium (Guillard’s nutrient supplement) and illumination with white fluorescent light (250 μmol photons m^−2^ s^−1^); 16:8 L:D for 8 h. The reported colour parameters (L*, a* and b*) of the powdered *P. palmata* biomass obtained were 52.19 ± 1.16, 3.58 ± 0.82 and 2.91 ± 0.40, respectively. In the study herein, *P. palmata* was cultured in the laboratory under an LED lighting system for a longer period i.e., up to 12 days, with a lower illumination of 100 µmol m^−2^ s^−1^ and at 10 °C which could have accounted for the differences in colour parameters between the two studies. For instance, the L* values (27.91–29.43) decreased, the a* values (7.23–8.04) increased and the b* values (6.09–7.99) increased following a 12-day culture of *P. palmata* under different light regimes in the present study. The colour data herein differed from our previous data on the impact of growth conditions (nutrients) during the culture of *P. palmata* [7]. In that study, the colour parameters of *P. palmata* cultured for 12 days at 10 °C with a photoperiod of 16:8 L:D at 100 µmol m^−2^ s^−1^ with the inclusion of different nutrient regimes such as algal nutrient supplement (F/2) alone or with urea (0.05 or 0.10 g/L) were measured. The colour parameters (L*, a*, b*) observed in the 12-day nutrient cultured samples with F/2 were 37.61 ± 0.62, 4.85 ± 0.29, 5.56 ± 0.46, respectively, with F/2 + 0.05 g/L urea were 36.20 ± 2.82, 4.86 ± 0.19, 4.91 ± 0.28, respectively, and with F/2 + 0.10 g/L urea were 35.50 ± 1.45, 4.41 ± 0.30, 5.00 ± 0.48, respectively. A decrease in L* value was observed when samples were cultured with F/2, F/2 + 0.05 g/L urea and the F/2 + 0.1 g/L urea, respectively. Similar a* values were observed when samples were cultured with F/2 and F/2 + 0.05 g/L urea and then a decrease in a* value for the sample cultured with F/2 + 0.10 g/L urea. Then, a decrease in b* value was observed for the sample cultured with F/2 + 0.05 in comparison with those cultured with F/2 and F/2 + 0.10, respectively.

In the study herein, the L* values decreased, a* values increased and b* values increased for *P. palmata* samples culture under different light spectra. The L* values decreased showing a similar pattern between the study and the current study while a different pattern was observed for the a* and b* values which could be due to variations in culture conditions. For instance, *P. palmata* was cultured with nutrient medium alone and with urea (0.05 or 0.10 g/L urea) in the study by Idowu et al. [7] while *P. palmata* was cultured with an algal nutrient medium with exposure to different light systems in this study. Exposure to different light regimes resulted in changes in the ΔE of the cultured *P. palmata* in the current study.

### 4.3. TPC of P. palmata Samples Cultured with Different Light Regimes

The study herein showed differences in the TPC of cultured *P. palmata* as a function of culture under different light systems and growth days. The highest TPC (11.21 ± 0.07 mg GAE/g dw) was observed in *P. palmata* cultured with red light on day 6 as shown in Figure 2. This sample also had the lowest TN (3.47 ± 0.03%), PN (2.36 ± 0.06%) and NPN (1.00 ± 0.06%) (Table 1). While samples cultured under red light for 12 days had a TPC of 8.18 mg GAE/g dw corresponding to TN, PN and NPN values of 3.87 ± 0.02%, 2.82 ± 0.05% and 0.97 ± 0.05%, respectively. On the contrary, samples cultured under blue light for 12 days had higher TN (4.73 ± 0.05%) and PN (3.56 ± 0.06%), respectively, which correspond with a TPC value of 5.91 ± 0.06 mg GAE/g dw. Phenolic compounds are secondary metabolites found in plants and other photosynthetic organisms (such as Phaeophyceae, which are not plants) that play a role in the defence against herbivores, pathogens and UV radiation, and which contribute to the colour of plants and seaweeds [66]. The presence of phenolic acids, flavonoids, phlorotannins and bromophenols in red seaweeds has been associated with their antioxidant activity [67]. Lomartire et al. [68] reviewed the phenolic content of seaweed and concluded that it decreased when seaweed was accumulating nutrients for biomass production; this may help explain the opposing tendencies between the PN and TPC, as observed herein. Therefore, accumulation of phenolic compounds may occur when seaweeds are not accumulating protein. This was illustrated in the case of *P. palmata* samples cultured under red light which had high TPC and relatively low TN and PN values. This is in contrast with samples cultured under blue and green lights which had lower TPC and higher TN and PN values. Hence, the potential exists to culture *P. palmata* under red light to enhance their phenolic content. Furthermore, seaweed phenolic compound accumulation can be triggered under stress conditions and during adjustment to various cellular metabolic activities [68]. This may account for the differences in TPC of the initial sample when compared to culture under white (1 and 2) light as seen by their different TPCs irrespective of their duration of growth.

A comparison of wild harvested and cultured *P. palmata* biomass grown with a nutrient medium in tanks at 12 °C at pH 8.0 in Nova Scotia, Canada and subsequently extracted with 80% acetone had TPC values of 4.8 and 5.5 mg GAE/g dw, respectively [69]. However, it has also been reported that the TPC content observed in macroalgae may be dependent on the extraction method (e.g., hot water, alcohol and organic solvents) used [69]. The use of organic solvents such as acetone, ethanol, ethyl acetate and methanol or a mixture of these organic solvents with water are mostly employed during the extraction of phenolic compounds from seaweed [70] due to their ability to enhance the solubilization of seaweed phenolic compounds [71]. Acetone and methanolic solutions were used for the extraction of the phenolic compounds from cultured *P. palmata* samples in the current study.

Wild harvested biomass from *Gracilaria edulis* obtained from India and subsequently extracted with 100% methanol had a TPC value of 3.4 ± 0.21 mg GAE/g [72]. Arulkumar et al. [73] reported the TPC of wild-harvested *G. corticata* also obtained from India to be 4.00 ± 0.35 mg GAE/g following extraction with 70% methanol. Ganesan et al. [74] reported a TPC of 16.26 mg GAE/g in *Gracilaria edulis* obtained from the west coast of India on extraction using 90:10 methanol:chloroform (*v*/*v*). Apart from the solvent used for extraction, the amount of phenolic compounds in seaweed is affected by storage conditions, source, sample particle size, purification techniques employed during sample preparation and the presence of interfering compounds in extracts such as pigments or fatty acid [68,75].

The study herein showed that the TPC of the *P. palmata* biomass varied with the different light regimes used during culture and with the duration of growth, factors which should be taken into consideration if the aim is to enhance the TPC in *P. palmata*.

### 4.4. Correlation between TN, PN, NPN and ΔE

A negative correlation between TN and ∆E seems likely that exposure to different light to increase TN and accumulate colour pigments could not be predicted despite a decrease in TN that was observed on day 6 of culture the TN values which subsequently increased following 12 days of culture (Table 1). This implies that there was no relationship between the accumulation of TN and colour change due to the accumulation of pigmented compounds. This may be attributed to differences in the TN and the colour parameters of the original sample vs. cultured sample, and the extent of nitrogen accumulation as a function of the different light regimes used during the culture of the samples [7].

The negative correlation between PN and ∆E analysis indicated a non-statistically significant negative correlation between the PN and the ΔE. This was as observed and stated above for TN vs. ΔE. The negative correlation between TN, PN and TPC showed that no correlation or relationship exists.

However, NPN indicated a positive correlation with TPC. It was previously shown that there were significant correlations between TN and the ΔE (r = 0.945, P = 0.016), between PN and the ΔE (r = −0.944, *p* = 0.001) and between NPN and ΔE (r = 0.734, *p* = 0.038) for cultured *P. palmata* with nutrient media (F/2) without and with urea (0.05 or 0.1 g/L) [7]. In the current study, *P. palmata* biomass was cultured with nutrient media (F/2) without urea under different light systems. In the study by Idowu et al. [7], *P. palmata* biomass was cultured with nutrient media (F/2) alone and with or without urea (0.05 or 0.1 g/L). The addition of urea (0.05 or 0.1 g/L) most likely enhanced nitrogen and colour pigment accumulation. This is the first study that reported on the Pearson correlation (if any) between TN, PN, NPN and ΔE as well as the relationship between TN, PN and NPN and TPC following culture of *P. palmata* in the laboratory under different light regimes, i.e., white 1, red, blue, green and white 2 during growth over 6 and 12 days.

### 4.5. SDS-PAGE Analysis

There appeared to be no distinct visible differences in the intensity of the bands despite the growth of the samples under different light spectra over 12 days. The reason for the lack of visualization of more intense bands in the stain-free gel particularly in the day 12 blue light culture, (Table 1) which had a higher TN, may be related to the modification of tryptophan residues when covalently bonded with trihalo compounds contained in the gel as described by Raynal et al. [76]. This likely accounted for the apparently lower intensity of protein band constituents particularly in the day 12 blue light culture.

Furuta et al. [77] reported that the presence of a band with a MW around 15 kDa corresponded to the α and β subunits of the phycobiliproteins in *P. palmata*. This band corresponds to band 11 (Figure 3) in the study herein. Other studies identified the subunit molecular weight of the phycobiliproteins to range between 16–18 kDa [78] and 30–33 kDa [79], which coincide with bands 11, 10 and 6 in this study. Galland-Irmouli et al. [80] reported 3 bands with molecular masses of 30, 21, and 20 kDa in wild harvested *P. palmata* which correspond to bands 9, 8 and 7 in the present study. Stévant et al. [81] reported a protein profile with molecular masses of 95, 72, 55, and 34 in wild harvested *P. palmata* which correspond to bands 2, 4 and 6 in the current study. Idowu et al. [7] identified seven protein bands with molecular masses corresponding to 55, 48, 33, 26, 20, 18 and 10 kDa following nutrient addition from culture medium alone and with urea (0.05 or 0.10 g/L) during culture of *P. palmata* samples which correspond to band 4, 6, 7, 9 and 10 in the study herein. This appears to be the first study on the SDS-PAGE protein profiles of *P. palmata* cultured under different light regimes over 6 and 12 days.

Samples cultured with different light regimes over a growth period of 6 and 12 days showed variation in % band volume in comparison to that of the initial sample. The consistency of the % relative band volume for bands 1, 2, 4, 5, 6, 7 and 8 indicates the presence of constitutive proteins whose expression was not influenced by the light system used during culture. Idowu et al. [7] recently reported on densitometry analysis of *P. palmata* proteins cultured with nutrient media (F/2) alone and with or without urea (0.05 or 0.1 g/L) on a stain-free SDS-PAGE gel. However, this appears to be the first report on densitometry analysis of *P. palmata* proteins obtained after culture under different light spectra.

### 4.6. AA Profiles

The predominant AA were Asx (combined aspartic acid and asparagine) and Glx (combined glutamic acid and glutamine) and they both constitute between 22.01–29.58% of the total AA (TAA). According to Stevant et al. [82], Asx and Glx are known to contribute to the umami flavour and taste of seaweed. Glx and Asx represented 23.1, 24.4 and 24.6% of the total AA (TAA) in wild-harvested *P. palmata* culture as reported by Aasen et al. [83], Mohammed et al. [84] and Bikker et al. [85] respectively. These AAs were also predominant constituting 23.8 and 12.6% of the TAA in other red seaweed such as *Pyropia* sp. [83] and *Chondrus crispus* [86], respectively.

It appears that culturing under different light systems did not alter the EAA composition of the biomass herein. Limited information appears to exist on the TAA and EAA profiles of intact *P. palmata* biomass cultured under different light regimes. A study by Idowu et al. [7] reported the TAA profile of *P. palmata* cultured for 18 days without nutrient media vs. with nutrient medium (F/2 and 0.05 g/L urea) at an illumination of 100 µmol m^−2^ s^−1^ and photoperiod of 16:8 (L:D) to be 3.12 ± 0.65 g/100 g vs. 18.98 ± 5.75 (g/100 g dw). The study indicated the possibility of increasing the TAA profile of cultured *P. palmata* using the nutrient medium. Mohammed et al. [84] reported the TAA of wild-harvested *P. palmata* to be 16.05 g/100 g dw.

Wild-originated *P. palmata* from the western seashore of Scotland and Ireland contained EAAs totalling 26.7 g/100 g dw (32.4% of the TAA) [85] and 5.92 g/100 g dw (36.9% of the TAAs) [84]. In this study, the culture conditions were controlled, e.g., the temperature maintained the metabolic activities and different light regimes that could possibly alter phycobiliprotein accumulation, as compared to the wild harvested samples obtained from west coast of Scotland and Ireland which rely solely on natural light and seawater nutrients for growth.

The NEAA values constituted between 54 to 65% of the TAA in the current study. Meanwhile, Mohammed et al. [84] reported NEAA values in wild-harvested *P. palmata* of 10.13 g/100 g dw (63.1% of the TAAs). The branched-chain AA (BCAAs) consisting of valine, leucine and isoleucine are important in nutrition because they play a key function in gut health, metabolism and energy homeostasis [87]. The use of different lighting regimes did not increase the TAA on days 6 and 12 except for the biomass cultured with blue light obtained on day 12 (27.98 ± 2.61, g/100 g dw). Further investigations may be warranted to explore the culture of *P. palmata* under blue light in order to enhance its AA profile. Mohammed et al. [84] reported the TAA of *P. palmata* to be 16.05 g/100 g dw for wild-harvested biomass. In the study herein, *P. palmata* cultured under blue light and grown for 12 days had the highest TAA value, i.e., 27.98 ± 2.61 g/100 g dw. Variation in TAA could be due to differences in biomass source (wild vs. culture), season of harvest (winter vs. summer), and the protocol followed for determination of amino composition as reported by Černá [88]. No study to date appears to report on the AA profile of *P. palmata* cultured under different light regimes and as a function of growth days.

In this study, variation existed in EAA, NEAA, BCAA and the TAA profiles of *P. palmata* cultured under different light regimes in comparison to the initial sample. Tryptophan was not determined due to its destruction during acid hydrolysis. However, the tryptophan content in wild-harvested *P. palmata* biomass harvested from the seacoast of Bodo, Norway was reported to be 0.27 g/100 g dw [89].

Different population cohorts require different amounts of EAAs. Therefore, it is recommended that the quality of a food protein should be quantified based on the above selected reference AA scoring patterns [41]. The study herein showed that the sulphur amino acid (SAA) score of the cultured *P. palmata* ranged from 1.97 to 14.49 for children from 6 months to 3 years and from 2.31 to 17.00 for older children, adolescents and adult population cohorts, respectively (Figure 5). According to the FAO [41], the AAS of a test food sample should be above 1 in all population cohorts in order to meet their dietary requirements. The study herein showed that the SAA content of the *P. palmata* cultured under different light regimes meets the levels recommended for the two population cohorts. The levels of threonine, valine, isoleucine, leucine, AAA, lysine and histidine all met the EAA requirement for the two population cohorts. Protein-rich food from other sources or AA supplementation would be required to enhance the limiting amino acids observed in the two population groups i.e., (i) children from 6 months to 3 years and (ii) in the older children older children (older than 3 years old), adolescents and adult population cohorts).

Overall, the culture of *P. palmata* with different light regimes (white 1, red, blue, green and white 2) for 6 and 12 days all displayed a good protein quality based on their AAS. These results demonstrate the possibility of enhancing the protein quality of *P. palmata* protein biomass through culture under defined light treatments. No previous reports appear to exist investigating the AA profile and score of *P. palmata* cultured under different light regimes.

## 5. Conclusions

From the results obtained for TN, PN, NPN, colour, TPC and AA profiles herein, it is evident that the specific light spectra used for culture impact the composition of *P. palmata*. The different light treatments impacted the overall nitrogen composition, colour and TPC along with the AA composition and AAS. Overall, culturing *P. palmata* under a blue light regime for 12 days appears to show promise in terms of enhancing its protein content when compared to the starting sample. Further work on pigment analysis is recommended in order to correlate pigment composition with protein content and colour. The results obtained herein provide new insights into how culture under different light regimes can impact nutrient accumulation during the growth of *P. palmata*. This information may be relevant when considering the large-scale cultivation of *P. palmata* in order to enhance the protein content and quality of this macroalgal species as a protein source for human consumption.

## Figures and Tables

**Figure 1 foods-12-03940-f001:**
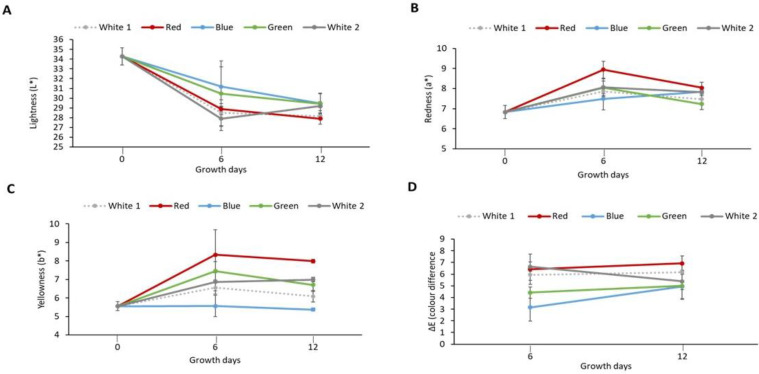
Colour parameters of laboratory cultured *Palmaria palmata* when grown over 12 days using different light regimes. Whole fronds of *P. palmata* were grown at a density of 4 g/L with a 16:8 Light:Dark photoperiod at 100 µmol/m^2^/s photosynthetic active radiation (PAR) with fertiliser (F/2) from different light-emitting diode sources: white (1 and 2), red, blue and green light using a Phillip’s Growise™ Control System for 6 and 12 days. (**A**) represents lightness (L*), (**B**) redness (a*), (**C**) yellowness (b*) and (**D**) colour difference (∆E) for laboratory grown *Palmaria palmata* under different light conditions over a period of 12 days. Data presented represents mean ± SD (*n* = 3). Initial sample was used as reference to calculate colour difference (ΔE) of the cultured *Palmaria palmata*.

**Figure 2 foods-12-03940-f002:**
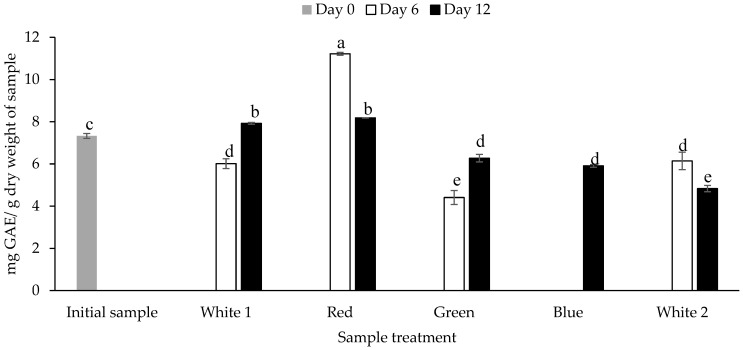
Total phenolic content (mg gallic acid equivalents (GAE)/g dry weight of sample) of *Palmaria palmata* biomass cultured under different light regimes. Whole fronds of *P. palmata* were grown at a density of 4 g/L with a 16:8 Light:Dark photoperiod at 100 µmol/m^2^/s photosynthetic active radiation (PAR) with fertiliser (F/2) from different light-emitting diode sources: white (1 and 2) red, green and blue light using a Phillip’s Growise™ Control System for 6 and 12 days. Values plotted are mean ± SD, *n* = 3. Different letters denote statistically significant differences (*p* < 0.05). Blue day 6 was not analysed due to a limitation in sample availability.

**Figure 3 foods-12-03940-f003:**
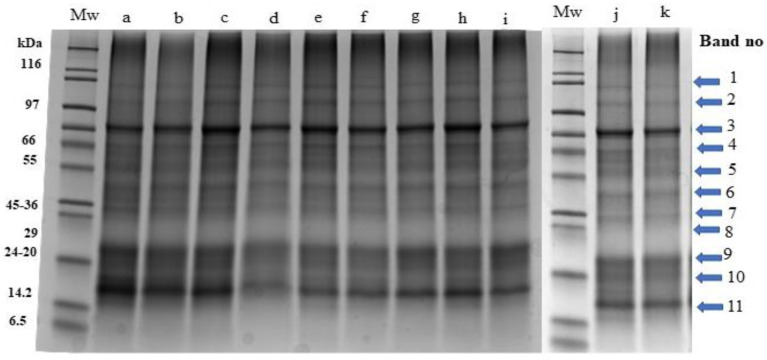
Stain free sodium dodecyl sulphate polyacrylamide gel electrophoresis profiles of *Palmaria palmata* protein extracts obtained from biomass grown under different colour light-emitting diode lights. Numbered arrows indicate the different protein bands within the samples. All lanes were loaded with 15 ug protein. Mw: molecular weight markers, a: Day 0-initial sample, b: Day 6-white 1, c: Day 6-red, d: Day 6-blue, e: Day 6-green, f: Day 6-white 2, g: Day 12-white 1, h: Day 12-red, i: Day 12-blue, j: Day 12-green, k: Day 12-white 2.

**Figure 4 foods-12-03940-f004:**
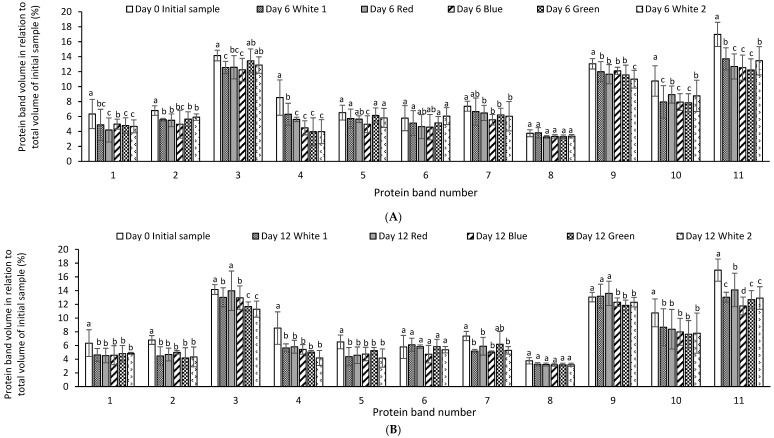
Comparative individual protein band intensity obtained using densitometry analysis of stain free sodium dodecyl sulphate polyacrylamide gel electrophoresis analysis of *Palmaria palmata* protein extracts originating from Galway and subsequently cultured for 6 (**A**) and 12 days (**B**) at 10 °C using different colour light treatments. Different letters denote significant differences within each protein band for the different light regimes and growth periods (*p* < 0.05). Bars are mean ± SD. Different letters indicate statistically significant differences within a given protein band (*p* < 0.05).

**Figure 5 foods-12-03940-f005:**
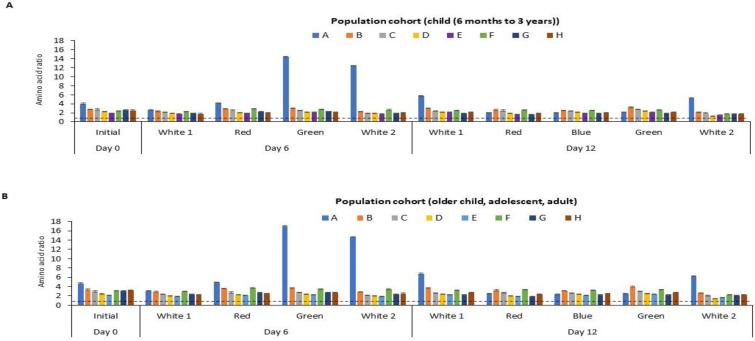
Calculated amino acid (AA) scores for essential amino acids for different age categories for *P. palmata* biomass cultured under different colour light conditions for a period of 6 and 12 days at 10 °C. Different AA residues/groupings are represented by capital letters, where A: sulphur amino acids (SAA), B: threonine, C: valine, D: isoleucine, E: leucine, F: aromatic amino acids (AAA), G: lysine, H: histidine. Sulphur amino acid: cysteine + methionine, Aromatic amino acid: tyrosine + phenylalanine. Tryptophan was not determined (nd). AA score was calculated as mg of AA per g of protein divided by the mg of the same AA per g of the reference protein for the essential AAs. Values are mean ± SD, (*n* = 2). Error bars represent 95% confidence limits. The dotted line depicts an AA score of 1.0. An AA score > 1.00 indicates that it meets the dietary requirement for the specific population cohort. An AA score < 1 is an indication that it does not meet the dietary requirement for the population cohort. The two selected population cohorts considered were (**A**) children (from 6 months to 3 years old) and (**B**) older children (older than 3 years old), adolescents and adults, according to the Food and Agriculture Organisation.

**Table 1 foods-12-03940-t001:** Total nitrogen (TN), protein nitrogen (PN) and non-protein nitrogen (NPN) content of *Palmaria palmata* biomass cultured under different light treatment conditions. Whole fronds of *P. palmata* were grown at a density of 4 g/L with fertiliser (F/2) and with a 16:8 Light:Dark photoperiod at 100 µmol/m^2^/s photosynthetic active radiation (PAR) provided by different light-emitting diode sources: white (1 and 2), red, blue and green light using a Phillip’s Growise™ Control System for 6 and 12 days.

	Light Details	TN (%)	PN (%)	NPN (%)
Day 0	Initial sample	4.56 ± 0.04 ^b^	3.20 ± 0.03 ^b^	1.21 ± 0.07 ^a^
Day 6	White 1	3.75 ± 0.05 ^f^	2.60 ± 0.04 ^d^	0.98 ± 0.03 ^b^
	Red	3.47 ± 0.03 ^g^	2.36 ± 0.06 ^f^	1.00 ± 0.06 ^b^
	Blue	3.71 ± 0.04 ^f^	2.48 ± 0.00 ^ef^	1.02 ± 0.06 ^b^
	Green	3.83 ± 0.05 ^e^	2.72 ± 0.07 ^c^	0.93 ± 0.01 ^b^
	White 2	3.49 ± 0.05 ^g^	2.42 ± 0.06 ^f^	0.99 ± 0.04 ^b^
Day 12	White 1	4.20 ± 0.06 ^d^	3.06 ± 0.06 ^b^	0.98 ± 0.02 ^b^
	Red	3.87 ± 0.02 ^e^	2.82 ± 0.05 ^c^	0.97 ± 0.05 ^b^
	Blue	4.73 ± 0.05 ^a^	3.56 ± 0.06 ^a^	1.01 ± 0.01 ^b^
	Green	4.39 ± 0.06 ^c^	3.15 ± 0.05 ^b^	1.06 ± 0.05 ^b^
	White 2	3.85 ± 0.05 ^e^	2.77 ± 0.06 ^c^	0.95 ± 0.07 ^b^

Values are mean ± SD, *n* = 3. Different superscript letters within each column denote statistically significant differences (*p* < 0.05).

**Table 2 foods-12-03940-t002:** Amino acid profiles (AA, g/100 g dry weight) of *P. palmata* biomass cultured under different colour light treatment conditions. Whole fronds of *P. palmata* were grown at a density of 4 g/L with fertiliser (F/2) and with a 16:8 Light:Dark photoperiod at 100 µmol/m^2^/s photosynthetic active radiation (PAR) provided by different light-emitting diode sources: white (1 and 2), red, blue, green light using a Phillip’s Growise™ Control System for 6 and 12 days.

Amino Acid	Day 0	Day 6				Day 12				
	Initial Sample	White 1	Red	Green	White 2	White 1	Red	Blue	Green	White 2
Asp	3.09 ± 0.83 ^a^	2.38 ± 0.77 ^b^	2.39 ± 0.32 ^b^	3.05 ± 1.18 ^a^	2.19 ± 0.13 ^b^	3.49 ± 0.25 ^a^	2.83 ± 0.36 ^ab^	3.27 ± 0.41 ^a^	3.03 ± 0.28 ^a^	2.66 ± 0.42 ^b^
Thr	1.27 ± 0.34 ^a^	0.88 ± 0.39 ^b^	1.00 ± 0.13 ^a^	1.19 ± 0.44 ^a^	0.80 ± 0.07 ^b^	1.33 ± 0.03 ^a^	1.07 ± 0.14 ^a^	1.30 ± 0.20 ^a^	1.47 ± 0.50 ^a^	0.86 ± 0.24 ^b^
Ser	1.67 ± 0.50 ^a^	1.21 ± 0.40 ^a^	1.30 ± 0.15 ^a^	1.60 ± 0.61 ^a^	1.08 ± 0.36 ^ab^	1.67 ± 0.20 ^a^	1.29 ± 0.15 ^a^	1.74 ± 0.27 ^a^	1.56 ± 0.02 ^a^	1.43 ± 0.29 ^a^
Glu	3.33 ± 0.89 ^a^	2.89 ± 0.74 ^b^	3.50 ± 0.49 ^a^	4.01 ± 1.53 ^a^	3.62 ± 0.22 ^a^	3.94 ± 0.09 ^a^	3.35 ± 0.38 ^a^	3.65 ± 3.76 ^a^	3.62 ± 0.26 ^a^	3.29 ± 0.45 ^a^
Pro	1.83 ± 0.53 ^c^	2.12 ± 0.66 ^b^	2.54 ± 0.40 ^b^	2.85 ± 1.22 ^b^	2.57 ± 0.11 ^b^	1.69 ± 0.26 ^c^	2.56 ± 0.34 ^b^	1.90 ± 0.35 ^c^	2.87 ± 0.30 ^b^	3.27 ± 0.53 ^a^
Gly	1.77 ± 0.53 ^a^	1.29 ± 0.37 ^a^	1.32 ± 0.22 ^a^	1.58 ± 0.63 ^a^	1.30 ± 0.11 ^a^	1.71 ± 0.05 ^a^	1.24 ± 0.19 ^a^	1.55 ± 0.04 ^a^	1.65 ± 0.33 ^a^	1.51 ± 0.20 ^a^
Ala	2.02 ± 0.65 ^a^	1.58 ± 0.47 ^a^	1.55 ± 0.24 ^a^	1.97 ± 0.80 ^a^	1.54 ± 0.06 ^a^	2.14 ± 0.05 ^a^	1.47 ± 0.20 ^a^	1.96 ± 0.07 ^a^	1.98 ± 0.22 ^a^	1.87 ± 0.26 ^a^
Cys	0.59 ± 0.18 ^c^	0.31 ± 0.17 ^c^	0.43 ± 0.08 ^c^	1.64 ± 0.75 ^a^	1.07 ± 0.11 ^ab^	1.58 ± 0.08 ^a^	0.23 ± 0.03 ^c^	0.30 ± 0.04 ^c^	0.26 ± 0.04 ^c^	1.14 ± 0.27 ^ab^
Val	1.76 ± 0.56 ^a^	1.14 ± 0.52 ^a^	1.21 ± 0.18 ^a^	1.42 ± 0.60 ^a^	0.95 ± 0.22 ^b^	1.47 ± 0.23 ^a^	1.42 ± 0.29 ^a^	1.71 ± 0.04 ^a^	1.76 ± 0.27 ^a^	1.08 ± 0.22 ^ab^
Met	1.04 ± 0.20 ^c^	0.57 ± 0.35 ^d^	0.83 ± 0.06 ^d^	3.36 ± 1.02 ^a^	2.78 ± 0.51 ^b^	0.65 ± 0.11 ^d^	0.52 ± 0.06 ^d^	0.59 ± 0.05 ^d^	0.57 ± 0.02 ^d^	0.73 ± 0.14 ^d^
ile	1.09 ± 0.36 ^a^	0.72 ± 0.38 ^b^	0.75 ± 0.12 ^b^	0.90 ± 0.37 ^ab^	0.68 ± 0.06 ^b^	1.01 ± 0.11 ^a^	0.81 ± 0.06 ^ab^	1.17 ± 0.02 ^a^	1.13 ± 0.02 ^a^	0.56 ± 0.14 ^b^
Leu	1.94 ± 0.64 ^a^	1.40 ± 0.60 ^ab^	1.43 ± 0.22 ^ab^	1.77 ± 0.74 ^a^	1.32 ± 0.08 ^ab^	2.01 ± 0.08 ^a^	1.49 ± 0.09 ^ab^	2.17 ± 0.07 ^a^	2.12 ± 0.06 ^a^	1.33 ± 0.36 ^ab^
Tyr	0.64 ± 0.21 ^a^	0.50 ± 0.07 ^a^	0.72 ± 0.03 ^a^	0.70 ± 0.17 ^a^	0.75 ± 0.11 ^a^	0.74 ± 0.20 ^a^	0.64 ± 0.08 ^a^	0.42 ± 0.10 ^a^	0.32 ± 0.06 ^a^	0.17 ± 0.12 ^b^
Phe	1.28 ± 0.40 ^a^	0.96 ± 0.31 ^ab^	0.95 ± 0.17 ^ab^	1.14 ± 0.45 ^a^	0.83 ± 0.17 ^b^	1.17 ± 0.24 ^a^	1.21 ± 0.15 ^a^	1.80 ± 0.07 ^a^	1.70 ± 0.04 ^a^	1.04 ± 0.07 ^ab^
His	0.77 ± 0.25 ^a^	0.44 ± 0.20 ^a^	0.45 ± 0.06 ^a^	0.56 ± 0.24 ^a^	0.45 ± 0.04 ^a^	0.62 ± 0.03 ^a^	0.49 ± 0.05 ^a^	0.66 ± 0.01 ^a^	0.63 ± 0.01 ^a^	0.47 ± 0.12 ^a^
Lys	2.26 ± 0.72 ^a^	1.38 ± 0.54 ^b^	1.43 ± 0.20 ^b^	1.69 ± 0.71 ^b^	1.26 ± 0.22 ^b^	1.57 ± 0.23 ^b^	1.19 ± 0.06 ^b^	1.74 ± 0.06 ^b^	1.60 ± 0.03 ^b^	1.32 ± 0.25 ^b^
Arg	1.31 ± 0.41 ^a^	0.81 ± 0.37 ^b^	0.86 ± 0.16 ^b^	1.16 ± 0.48 ^a^	0.88 ± 0.08 ^b^	1.28 ± 0.07 ^a^	0.71 ± 0.06 ^b^	1.18 ± 0.06 ^a^	1.09 ± 0.02 ^a^	0.88 ± 0.24 ^ab^
Trp	nd	nd	nd	nd	nd	Nd	nd	nd	nd	nd
ƩEAA	12.71 ± 3.88 ^a^	10.11 ± 2.66 ^b^	8.92 ± 1.30 ^b^	10.33 ± 0.77 ^b^	9.23 ± 1.33 ^b^	11.11 ± 0.10 ^b^	9.29 ± 0.64 ^b^	12.51 ± 0.58 ^a^	12.06 ± 0.81 ^a^	8.26 ± 0.10 ^bc^
ƩNEAA	14.94 ± 3.97 ^b^	13.84 ± 3.48 ^c^	13.74 ± 1.87 ^c^	13.53 ± 1.63 ^c^	13.54 ± 1.20 ^c^	16.97 ± 1.19 ^a^	14.56 ± 1.70 ^b^	15.47 ± 2.04 ^b^	15.30 ± 1.37 ^b^	15.34 ± 0.85 ^b^
ƩBCAA	4.79 ± 1.56 ^a^	4.01 ± 1.06 ^ab^	3.39 ± 0.52 ^b^	3.11 ± 0.46 ^b^	2.75 ± 0.34 ^c^	4.49 ± 0.78 ^a^	3.96 ± 0.42 ^ab^	5.10 ± 0.13 ^a^	5.01 ± 0.34 ^a^	2.97 ± 0.03 ^c^
ƩTAA	27.65 ± 7.85 ^a^	23.94 ± 6.14 ^b^	22.66 ± 3.16 ^c^	23.86 ± 2.40 ^b^	22.77 ± 2.53 ^c^	27.50 ± 1.27 ^a^	23.85 ± 2.33 ^b^	27.98 ± 2.61 ^a^	27.37 ± 2.18 ^a^	23.60 ± 0.96 ^b^

All amino acids are represented by their 3-letter code. EAAs: essential amino acids, NEAA: non-essential amino acids, BCAA: branched chain amino acids, Tryptophan was not determined (nd) since it was destroyed during acid hydrolysis. Glx (glutamate and glutamine) and Asx (aspartate and asparagine) are reported since acid hydrolysis converts glutamine and asparagine to their corresponding acids. Values represent mean ± SD (*n* = 3). Superscript letters represent significant differences within a row (*p* < 0.05). The amino acid profile of blue day 6 was not analysed due to a limitation in sample availability.

## Data Availability

The data used to support the findings of this study can be made available by the corresponding author upon request.

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
