# Peer review of "Impact of Different Light Conditions on the Nitrogen, Protein, Colour, Total Phenolic Content and Amino Acid Profiles of Cultured *Palmaria palmata"

_foods, 2023, doi:10.3390/foods12213940_

Round 1

Reviewer 1 Report

Comments and Suggestions for Authors

The study investigated the impact of culturing the red seaweed Palmaria palmata under different LED light colors (white, red, blue, green) for 6 and 12 days on its nitrogen, protein, color, total phenolic content (TPC), and amino acid profiles.

  • The study addresses an important research gap regarding the effects of different LED light colors on the nutritional composition of cultivated Palmaria palmata. Very few studies have examined this.
  • The results are novel and reveal some interesting effects of light color, especially the potential of blue light to increase protein content and red light to increase phenolics. This could inform optimal cultivation strategies.

I strongly recommend the following modifications:

  • The lack of pigment analysis (chlorophyll, carotenoids, phycobiliproteins) makes the color and composition results more difficult to interpret mechanistically.
  • More details needed on the cultivation system parameters (mixing, aeration, nutrients etc.) to aid reproducibility.
  • Figure 5 needs statistical analysis.

Overall the study design and methods are appropriate and the results contribute some useful new knowledge. However, biological replication should be improved in future studies to substantiate the conclusions. Some additional measurements and details would aid interpretation and reproducibility. The findings represent an important starting point for further optimization studies.

Author Response

see attached file response to reviewers

Reviewer 2 Report

Comments and Suggestions for Authors

This is an interesting paper regarding the impact of different LED light colours (white, red, blue and green) on the chemical composition of cultivated Palmaria palmata under controlled laboratory-scale conditions. Being one of the few authorized edible seaweeds in Europe, optimizing its cultivation is of utmost importance to promote its consumption, production and trade.

The authors prove that manipulation cultivation conditions (nitrogen and light quality and intensity indeed change seaweeds’ biomass quality, namely nitrogen and thus increasing protein content.

Introduction

The introduction contains information relevant to the subject researched, the objective, and the importance of the study are well-defined and highlighted. Reference is made to some previous results obtained by other authors, and the justification for the current experiment is well argued in the context of these preliminary works.

It should be noted that the species Palmaria palmata is one of the few macroalgae species approved by the EU's Novel Food Catalogue. https://food.ec.europa.eu/safety/novel-food_en

Methods:

The research methodology is presented with precision.

I agree with the use of a 4.7 conversion factor for this red seaweed. These always have much lower efficiency converting nitrogen-to-protein than the usual 6.25 conversion factor, overestimating the protein content.

Protocols usually state acclimatization of 7 days (one week). You had only four days. Can you explain?

I wonder why the experiments took only 12 days. Because you were working with seaweeds, I believe a longer trial period would have a more significant effect on the biomass.

Results:

The experimental results are clearly presented and are easily perceptible.

line. 256. Because this is the beginning of a new section of the paper, refresh the TN, PN, NPN meaning (Total nitrogen, protein nitrogen and non-protein nitrogen). It has been a while since they have been defined.

Line. 292 – P value is a lowercase letter

Figure 2 – number of samples is missing.

Figure 3 – because the quality of the pictures is not very good: the lanes are not straight, so and the bands are not clearly indicated, either you improve the figure or you remove it.

3.6. AA profiles.

Full name for BCAA (Branched Chain Amino Acids)

Discussion:

The discussion of the results has been well elaborated and is supported by the introduction which defines the purpose of the research. Logical and coherent assumptions are issued, and the directions that need to be further researched are highlighted to elucidate issues that are not yet fully clarified.

In detail:

4.1.

 I believe that your results would have been different if you had longer acclimatization; a 4-day period is too little. The seaweeds were still stressed due to the different conditions in the lab. vr. field. I also agree that seaweeds adapt their phycobilisomes to light quantity and quality. However, they may take longer than 12 days to do so.

To confirm these thoughts, it would have been useful to measure phycoerythrin, as it promotes light harvesting, photosynthesis, and further metabolism.

Line 105. Chlorophyll and phycobiliproteins are quite different pigments. So I wouldn’t use “therefore”. I suggest “furthermore”.

4.2.

I believe you can correlate L*, a* and b* with pigment production by the seaweed. Higher phycoerythrin production will increase darkness (lower L*) and higher a*. This is consistent with the previous TN, PN results since phycobilisomes are proteins.  As to b* maybe you can correlate with carotenoids increase.

4.3.

Line 155 – phenolics occur in plants and other photosynthetic organisms (such as Phaeophyceae, which are not plants). So correct the sentence.

Conclusions:

The conclusions are concise, pointing out the main results obtained in the experimental study and emphasizing the importance of the subject addressed.

Author Response

see attached response to reviewers
